# Do Pictograms on Medication Packages Cause People to Consult Package Inserts Less Often? If so, With What Consequences?

**DOI:** 10.3390/bs13080696

**Published:** 2023-08-21

**Authors:** Ester Reijnen, Lea Laasner Vogt, Swen J. Kühne, Jan P. Fiechter

**Affiliations:** School of Applied Psychology, ZHAW Zurich University of Applied Sciences, Pfingstweidstrasse 96, CH-8005 Zurich, Switzerland; lea.laasner@zhaw.ch (L.L.V.); swen.kuehne@zhaw.ch (S.J.K.);

**Keywords:** pharmaceutical pictograms, information seeking, PI consultation, correct dosage, adherence

## Abstract

Overall, pharmaceutical pictograms seem to improve medication adherence. However, little is known about how warning pictograms (e.g., “do not drive after taking”) on medication packages influence patients’ information-seeking strategies such as consulting the package insert (PI) to determine other features such as the correct dosage. In this online study, participants (358 students) were presented with three fictitious scenarios (e.g., headache after alcohol consumption; factor scenario) in which medication use would be contraindicated. Each scenario was accompanied by a visual presentation of a medication package that could contain three possible pictogram selections or arrangements (factor warning); some arrangements contained pictograms relevant to the situation represented by the scenario, while others did not. Participants had to decide which dosage of the represented medication they were allowed to take in the given scenario. In making this decision, they could consult the PI or not. Overall, in two out of the three scenarios (driving and pregnancy), medication packages with relevant pictograms resulted in fewer PI consultations but led to more correct dosage decisions (“no pill”) than packages with irrelevant pictograms. Pictograms generally played no role in either the review of the PI consultation or dosage decisions in the alcohol scenario. Providing warning-relevant pictograms on medication packages can help people know when they should not take medication even without reading the PI.

## 1. Introduction

Imagine the following scenario: You have a terrible headache, but you still have several hours of driving ahead of you. Hence, you stop at the nearest pharmacy and buy a non-prescription or *over-the-counter (OTC)* pain reliever. The question is do you read the package insert (PI) to learn about possible side effects (e.g., dizziness) or warnings (e.g., “do not drive after taking”)? And what could be the consequences of not consulting the PI in regard to the correct dosage, for example? After all, what good is it if the headache disappears when you take the pain reliever, but you become a danger on the road?

One reason people do not read PIs is that they are far *too complex* and, accordingly, poorly understood. People of all educational levels struggle with this problem [1,2,3], as do those who do not speak the local language (e.g., immigrants and tourists). The consequence of this is medication *non-adherence*, that is, not taking medications as described in terms of timing, dosage, etc. [4]. Non-adherence leads not only to illness and death [5] but also to enormous healthcare costs [5,6]. In the US alone, these costs range from USD 100 to USD 290 billion annually [6]. 

Accordingly, in recent years, many efforts have been undertaken to improve the understanding of PIs by adding pharmaceutical pictograms [7,8,9,10]. Pharmaceutical pictograms are *visual aids* representing information regarding, for example, the use of medication (e.g., ear drops) [11]. Extensive research exists on the beneficial effects of pictograms [12,13]. Aptly summarized in a quote from Ros Dowse, pictograms are “more easily processed by the brain and are interpreted more accurately and rapidly than words or text and are also superior to text in prompting recall of information” [14] (p. 1518). The use of pictograms, therefore, improves medication adherence (e.g., in 59% of cases) [15], especially when pictograms are combined with text [2,13,16]. However, caution should also be exercised when using pictograms because they are context-dependent (e.g., on culture) [17]. For example, Wang and colleagues have shown that American participants better understand more complex and more concrete pictograms, while Chinese participants tended to better understand less complex and less concrete pictograms [17]. 

Nevertheless, given the promising results with the use of pictograms, researchers have recently begun testing whether the use of pictograms also has a beneficial effect when used on vials or medication packages [16,18,19], for example, by displaying the most serious side effects. However, little to nothing is known about how pictograms, especially warnings, on medication packages influence consultation of PIs. For example, do they promote reading or do they tend not to? More importantly, how do pictograms influence medication dosage choice directly or indirectly via PI consultation?

An important fact to take into consideration is that another reason PIs are not consulted is that people prefer not to learn (or know) about side effects or medication-related restrictions (e.g., driving). Accordingly, recent research has shown that people prefer to remain ignorant when it comes to gaining information about future outcomes [20], especially if the expected outcome is undesirable [21]. For example, 42% of individuals would not want to know whether they have a genetic predisposition to certain cancers [21]. When it comes to the potential side effects of pain relievers, 35% of participants prefer not to be informed [22]. 

This leads us to the question of how pictograms or visual aids (e.g., regarding warnings and/or side effects) interact with ignorance. To our knowledge, there are only a very limited number of such studies, but these are in other subject areas such as food. For example, Thunström and colleagues [23] have demonstrated that 58% of the participants in their treatment group (with no visual aids) chose not to know (i.e., stay ignorant) about the calorie content of their preferred meal (in a choice between a high-calorie chicken meal and a low-calorie beef meal). Subsequently, the ignorant participants consumed significantly more calories (626 kcal) than the ones that wanted to know (informed: 522 kcal). However, the informed participants consumed a similar amount of calories as the participants in the control group (532 kcal) who were informed in advance, visually and verbally, about the meal’s calories. Note that the calories displayed were calculated based on the participants’ final decision (i.e., revised decisions after receiving the information). Hence, when pictograms are shown on a medication package, the situation is somewhat like in the control group, as important information is conveyed right from the beginning. Therefore, pictograms on medication packages might be helpful in that they provide relevant information without people having to read the PI.

To return to our scenario, regardless of what the reason might be for not reading the PI (too complex or ignorance), it must be ensured that pictograms convey the same message as the PI. This requires that pictograms are processed to the extent that their identity or *meaning* is grasped (e.g., oh, a pictogram that tells me that I am not allowed to drive a car), which depends significantly on a “good” pictogram design [11,24]. In addition, pictograms—in order to be used in a targeted manner—must enable the decision as to whether or not the corresponding pictogram is relevant to the situation at hand (a process that is assumed to take place after identification; see [25]). In other words, if you still have several hours of car journey ahead of you, for example, a pictogram indicating that you should not drive a car after taking the medication is relevant. However, one that indicates “do not take if you are pregnant” would not be relevant if you are male. The question is whether pictograms can not only initiate these two processes (identification and decision making) but also bring them to completion. Incidentally, the final steps of the first process (i.e., identification) as well as the decision process require the use of “exhaustive” cognitive resources (i.e., attention). Therefore, pictograms might well be avoided or skipped.

In regard to displaying relevant information, we wonder whether the position or sequence of the pictograms displayed matters. More specifically, does the warning pictogram “do not drive after taking” placed in the first position have a differential effect compared with when it is in the second position? Studies have shown that participants typically read rows of information from left to right (in societies where the reading direction is from left to right) [26]. This means that information further to the left is not only seen (or attended to) first but also processed first. More importantly, studies on the spatial representation of magnitudes (e.g., time duration, number magnitude, and spatial extent) show that participants mentally map an increase in magnitude from left to right [27,28]. For example, participants would mentally map the healthfulness of food from healthy (left) to unhealthy (right). In that sense, Romero and Biswas [29], have shown that healthy foods (i.e., salad) are chosen almost twice as often when they are positioned to the left (vs. right) of unhealthy foods (i.e., burger). Chae and Hoegg [27] provide some examples outside the food sector, such as lamps. 

In conclusion, the present study investigates whether pharmaceutical pictograms displayed on medication packages influence PI consultation and the correct dosage choice. Given the existing literature, we hypothesize that pictograms circumvent ignorance and thereby improve dosage decisions even when participants do not consult the PI. In addition, we hypothesize that a warning pictogram in the second position (versus the first position) will be perceived as more serious or associated with more risk, which should lead to increased PI consultation and/or more correct dosage decisions.

## 2. Materials and Methods

### 2.1. Participants 

In total, 358 participants from the ZHAW (Zurich University of Applied Sciences) and the greater Zurich area took part in this computer-based online study. According to the local Ethics Committee, there were no ethical objections. The participants were recruited via the university’s internal student e-mail list. Their age ranged from 18 to 54 years of age (*M* = 25.7, *SD* = 6.1), 63% of whom were female. About three-quarters of the participants (71.2%) stated that their highest degree was a baccalaureate (e.g., specialized baccalaureate, federal vocational baccalaureate), 21.5% had a university degree, and 7.3% had any other degree or chose not to provide information about this factor. Participants were able to enter an iPad raffle. Students from the School of Applied Psychology (17.3%) had an additional option whereby they could receive course credit for their participation instead (54.8% of them did). All participants provided written informed consent. 

### 2.2. Stimulus Material 

For each of the three medications (one for each scenario; see Procedure and Design), we designed a corresponding medication package (an example is provided below in Figure 1). The package could contain three out of six possible pictograms in the lower right corner. Three of these pictograms contained a warning (i.e., “do not take during pregnancy”; “do not take in combination with alcohol”; and “do not drive after taking”). The other three pictograms contained possible side effects (i.e., “can cause tiredness”; “can cause nausea”; and ”can cause dizziness”; see Figure 2 for all the pictograms used). The pictograms were designed in cooperation with two designers and summarized under the name Pharmaceutical Pictogram System (PPS), which includes other pictograms. For each medication, there was also a fictitious PI (see Appendix A for an example in German). These were identical in terms of the text concerning the three warnings and three side effects.

### 2.3. Procedure and Design 

Each participant was presented with *three fictitious scenarios* (in random order) in which they either had to put themself into the situation that they (a) had drunk two bottles of beer in the last three hours and back pain appeared (alcohol scenario), or (b) still had a two-hour car ride ahead of them and had a headache (driving scenario), or (c) were seven months pregnant and had a migraine (pregnancy scenario); together, these constituted the “scenario” factor. In addition, participants were informed that, given their situation, they would visit a pharmacy, which would offer them an appropriate OTC medication (either for back pain, headaches, or migraines) to combat the pain described in the scenario. 

The medication or its packaging was presented visually to the participants. (The medication package with/without pictogram appeared together with the cover story [page 1] and again at the dosage question [page 2]. Thus, the package was always visible.) They could contain the following pictogram selections or arrangements (“warning” factor): (1) the relevant pictogram, that is, the relevant warning (e.g., “do not drive after taking”) in the first position combined with two possible side effects in the second and third position (this arrangement was called “first position”); (2) the relevant warning in the second position with another warning in the first position and a possible side effect in the third position (this arrangement was called second position); or (3) no warning at all but just three possible side effects (this arrangement was called none). 

The pictogram conditions were assigned to each scenario in such a way that the relevant pictogram was shown in the first position in one of the scenarios, the relevant pictogram was shown in the second position in another scenario, and no warning pictograms (i.e., only side effects) were shown in the remaining scenario. We used a Latin square to assign the different warnings to the scenarios per participant (see Appendix A for more detailed information). In addition, there was a separate control condition (the so-called fourth condition, hence also a condition of the factor “warning”) in which participants did not see pictograms for any scenario (see Appendix A regarding the number of participants per condition).

Crucially, regarding each scenario, participants had the task of indicating the dosage they would take (dependent variable “dosage”). The dosage options were 1 pill, ½ pill, ¼ pill, or no pill. Although the stated recommended dosage of medication was one tablet, the correct dosage under the to-be-imagined circumstances was “no pill” in all scenarios. Importantly, each participant was offered the additional option of consulting the PI first before making their decision (dependent variable “PI consultation”). If the participants chose this option, they were shown the PI. After reading it, they were again asked the dosage question. At the end of the study, the participants were asked about sociodemographic data (e.g., sex, age, etc.). 

## 3. Results

The participants who needed more than 30 or less than 1.5 min to complete the whole study (3%) were excluded from the analysis, leaving 358 participants. All calculations were performed with R software (version 2023.6.0.421, Boston, MA, USA) [30].

### 3.1. Consultation of the PI

To begin with, we were interested in whether the factors warning and scenario influenced participants’ willingness to consult the PI; if so, in what ways? We, therefore, calculated a probit regression model instead of a logit model (due to a lower number of participants in the control group, the calculation was fitted into an ANOVA) with the scenario (alcohol, driving, and pregnancy) and warning (first position, second position, none, and control) factors on the percentages of the binary outcome variable (did or did not consult the PI). We found a significant main effect scenario (χ^2^ (2, *N* = 1074; note that each of the 358 participants made three decisions each, so the total number of decisions was *N* = 1074) = 7.02, *p* < 0.05) and a significant main effect warning (χ^2^ (3, *N* = 1074) = 68.67, *p* < 0.001). Moreover, we also found a significant scenario × warning interaction (χ^2^ (6, *N* = 1074) = 49.70, *p* < 0.001) (see Figure 3).

We calculated two 3 (scenario: alcohol, driving, and pregnancy) × 2 (warning: first position versus second position or warning: none versus control) ANOVAs on the percentages of the binary outcome variable (did or did not consult the PI). There was no significant main effect for warning conditions of first position and second position (χ^2^ (1, *n* = 610) = 0.04, *p* = 0.84, respectively) or warning conditions of none and control (χ^2^ (1, *n* = 464) = 0.01, *p* = 0.91). In addition, no significant interaction was found in either analysis (χ^2^ (2, *n* = 610) = 2.09, *p* = 0.35 for warning: first position and second position, respectively; and χ^2^ (2, *n* = 464) = 1.90, *p* = 0.39 for warning: none and control). Thus, given that there was no significant difference between either the warning conditions “first” and “second” positions (which both included the relevant warning pictogram on the medication package depicted) or the warning conditions “none” and “control” (which both omitted the relevant warnings), for the remaining analyses, we collapsed the data to create a binary comparison between the relevant condition (i.e., the former warning conditions first and second positions) versus the irrelevant condition (i.e., the former warning conditions none and control). 

This also makes sense from a theoretical point of view, as it emphasizes the importance of displaying relevant pictograms over irrelevant ones. Hence, reanalyzing the data over the collapsed variables should not change the observed pattern. Indeed, as expected, we again found a significant main effect of scenario (χ^2^ (2, *N* = 1074) = 7.01, *p* < 0.05) as well as warning (χ^2^ (1, *N* = 1074) = 68.66, *p* < 0.001) factors; moreover, a significant scenario x warning interaction was also observed (χ^2^ (2, *N* = 1074) = 45.67, *p* < 0.001). Regarding this interaction, post hoc tests (Bonferroni-corrected) showed no significant difference in reading the PI between the relevant and irrelevant warning conditions in the alcohol scenario (*p* > 0.999). This contrasts with the significant differences in the driving and pregnancy scenarios, with both having *p* < 0.001; this difference was more pronounced in the pregnancy scenario. 

Importantly, overall, the PI was read more frequently in the irrelevant condition (48%) than in the relevant condition (23%). In the pregnancy scenario, only 16% of the participants consulted the PI in the relevant conditions—is this wise in terms of choosing the correct dosage? If the illustrated pictograms convey the intended information, then yes, otherwise no. In the next sections, we, therefore, consider the dosage decision first separately with and without PI consultation and then overall at the end. 

### 3.2. Dosage without PI Consultation

How did participants who did not consult the PI (66% overall) decide on dosage? To investigate this question, we calculated an ordinal logistic regression with the factors scenario (alcohol, driving, and pregnancy) and warning (relevant and irrelevant) on the ordinal-scaled data dosage (i.e., 1, ½, ¼, or no pill). It was found that both main effects, i.e., scenario (χ^2^ (2, *n* = 711) = 29.75, *p* < 0.001) and warning (χ^2^ (1, *n* = 711) = 86.74, *p* < 0.001), became significant. Furthermore, the scenario x warning interaction also became significant (χ^2^ (2, *n* = 711) = 14.27, *p* < 0.001) (see Figure 4). We again found no significant main effects for warning (first position and second position, respectively, none, and control) regarding dosage (both χ^2^-values < 1.39, both *p*-values > 0.24), nor any significant interactions (both χ^2^-values < 3.88, both *p*-values > 0.14). Accordingly, Figure 4a shows that, in the relevant warning condition, 71% of the participants made the correct choice (no pill), compared with only 35% in the irrelevant condition. Because of the small number of cases for some dosages (e.g., ¼ pill), the data were collapsed across the incorrect dosages (i.e., 1 pill, ½ pill, and ¼ pill). Subsequently, we again calculated a probit regression model with the two factors scenario and warning, but this time over the binary outcome (did or did not choose “no pill”). The pattern described above was confirmed: again, both main effects (scenario: χ^2^ (2, *n* = 711) = 37.01, *p* < 0.001 and warning: χ^2^ (1, *n* = 711) = 74.76, *p* < 0.001), as well as the scenario x warning interaction (χ^2^ (2, *n* = 711) = 17.65, *p* < 0.001), became significant (see Figure 4b). 

Post hoc tests (Bonferroni-corrected) confirmed this interaction pattern: With respect to the relevant warning condition, all scenario comparisons (alcohol vs. driving and alcohol vs. pregnancy, as well as driving vs. pregnancy) were significant (all *p*-values < 0.05), but not with respect to the irrelevant warning condition (all *p*-values > 0.999). Furthermore, post hoc tests (Bonferroni-corrected) showed that there was no significant difference between the relevant and irrelevant warning conditions in the alcohol scenario (*p* = 0.41), but there were significant differences in the driving and pregnancy scenario (both *p*-values < 0.001). Overall, it can be stated that the “no pill” decisions made without consulting the PI were more often accurate in the driving and pregnancy scenarios when participants were presented with the relevant pictograms.

### 3.3. Dosage with PI Consultation

How did participants (about 34%) decide when they were informed, i.e., when they consulted the PI? [Here, in contrast to before, we found significant main effects for warning (first position and second position: χ^2^ (1, *n* = 142) = 6.87, *p* < 0.01, respectively; none and control: χ^2^ (1, *n* = 221) = 7.60, *p* < 0.01) regarding dosage, but again no significant interactions (both χ^2^-values < 2.00, both *p*-values > 0.37). These results are discussed in Section 4)]. As can already be inferred from Figure 5a, the calculated ordinal logistic regression with the factors scenario (alcohol, driving, and pregnancy) and warning (relevant and irrelevant) on the ordinal-scaled data dosage (1, ½, ¼, or no pill) showed only a significant main effect of the scenario factor, with χ^2^ (2, *n* = 363) = 36.98, *p* < 0.001 (warning: χ^2^ (1, *n* = 363) = 0.48, *p* = 0.49; scenario x warning interaction: χ^2^ (2, *n* = 363) = 0.77, *p* = 0.68). For the reasons stated above, the data across the incorrect dosages (i.e., 1 pill, ½ pill, and ¼ pill) were once more collapsed. Again, the calculated probit regression model with the two factors scenario and warning over the binary outcome variable (did or did not take any pill; see Figure 5b) showed the same pattern as above: There was only a significant main effect of scenario (χ^2^ (2, *n* = 363) = 37.23, *p* < 0.001). Neither the warning main effect (χ^2^ (1, *n* = 363) = 0.75, *p* = 0.39) nor the scenario x warning interaction (χ^2^ (2, *n* = 363) = 2.95, *p* = 0.23) was significant. A comparison (Bonferroni-corrected) across both warning conditions showed that the alcohol and driving scenarios were not significantly different (*p* = 0.62), but that there was a significant difference between these two scenarios compared with the pregnancy scenario, both *p*-values < 0.001.

### 3.4. Dosage OVERALL

To examine the overall impact of providing pictograms, we again calculated a probit regression model over the full sample with the factors warning (relevant and irrelevant) and PI consultation (with and without) on the binary outcome variable (did or did not take any pill). The scenario factor was not considered. We found a significant main effect for warning (χ^2^ (1, *N* = 1074) = 42.42, *p* < 0.001) and also for PI consultation (χ^2^ (1, *N* = 1074) = 38.55, *p* < 0.001). More interestingly, we found a significant warning x PI consultation interaction (χ^2^ (1, *N* = 1074) = 49.52, *p* < 0.001) (see Figure 6). Post hoc tests (Bonferroni-corrected) confirmed this interaction pattern: The selection of “no pill” was similar in the relevant warning conditions regardless of whether the participant consulted the PI (*p* > 0.999). The rates of choosing “no pill” were also similar among those who chose to consult the PI, regardless of whether they were in the relevant warning or irrelevant warning conditions (*p* = 0.24). However, participants in the irrelevant warning condition who did not consult the PI differed from all the three other conditions or groups (all *p*-values < 0.001). 

## 4. Discussion

Overall, our study shows that the use of pictograms and the type of pictograms presented (i.e., relevant or not) influence whether people consult PIs and choose the “correct” medication dosage. More specifically, the rates of making the correct dosage choice (“no pill”) among the participants receiving relevant warnings were similar to those among participants who consulted the PI. Furthermore, our study shows that the number of people consulting the PI is relatively low (about one-third in our case; see Vinkers and colleagues [31] for a higher value of about 52%, or Orayj and colleagues [32] for a value of 65%). For this reason, the use of pictograms is all the more important as they may reduce the inappropriate use of medications in certain situations (e.g., driving). The difference in values across studies with respect to PI consultation could be due to what is known as the attitude–behavior gap, or more specifically, how the target behavior (here, PI consultation) was measured in each study (i.e., the intention–behavior gap): as *stated* behavior (whether collected through survey or laboratory data) or as *actual* behavior. In this context, in the food domain, Moser [33] found that environmental attitudes influence self-reported purchase behavior for organic food but not their actual purchase (although labels could be helpful here, see Vermeir et al. [34]). In the study by Orayj et al. [32], participants only had to indicate whether they would consult the PI, whereas in our study, we measured actual PI consultation. It should be noted, however, that the number of PI consultations in reality (e.g., due to the omission of social expectation) might be even lower than in our study. However, further research is needed in this area. 

Why might the use of pictograms on medication packages be beneficial? Once again, it proves useful to look at the food sector since it faces a similar problem. Until a few years ago, to decide how healthy a food product was, consumers had to consult the nutrition information (i.e., the so-called nutrition facts panel) on the back (or sometimes side) of a product. This is challenging to consumers, not only because of time constraints but also because of the complexity of the textual information. Consumers often have a hard time understanding the information [35,36]. Even if they understood the information, in terms of defining the healthiness of a product, one would have to carefully consider all the relevant facts and their respective significance (e.g., the amount of sugar, salt, etc.). This would require the use of some strenuous cognitive processing. Research by Kahneman [37] has shown that people prefer the simpler method of focusing on only one fact (e.g., sugar) and deciding on that basis (i.e., a lot of sugar = unhealthy; see also Reijnen et al. [38]) rather than using strenuous cognitive activity to consider all the relevant facts. To overcome the problems with nutrition facts panels, and to provide consumers with adequate information, the so-called front-of-pack labels have been developed. As the name suggests, these labels are placed on the front of the package and indicate in a simplified format whether the product is healthy or not [39]. The most prominent ones in use today are the traffic light label and the Nutri-Score. Research has shown that these front-of-pack labels are generally beneficial in improving consumers’ knowledge and helping them choose healthier products [40,41]. However, it is still debated whether front-of-pack labels can also have a positive impact on purchase intention and consumption. One reason for the conflicting results in this area so far could be that the studies were conducted in different settings (e.g., hospitals, restaurants, etc.), as well as in different continents/countries. Furthermore, different labels have shown different effects: in terms of purchase intention, the winner so far seems to be the traffic light label [42,43] (see for more unclear results). 

Similar positive (but also negative) implications may be indicated for pharmaceutical pictograms in place of, or in addition to, the more complex PIs. However, to develop their beneficial effects in the later stages (i.e., from comprehension to consumption or adherence), they need to be able to attract attention. This applies to both front-of-pack labels and pharmaceutical pictograms. If this is not the case, the earlier processing (e.g., perception) of relevant information is inhibited, and it is more difficult to reach the later processing stages (comprehension, decision, consumption, and adherence). In this regard, Becker et al. have shown that, compared with nutrition facts panels, front-of-pack labels are not only noticed earlier (2.7 s vs. 4.5 s) but also more frequently (98.3% vs. 31.4%) [44]. Furthermore, the front-of-pack effect was more pronounced if the labels contained color. To our knowledge, this has not yet been explored in the field of pharmaceutical pictograms. However, the authors of this paper have shown that their pictograms, developed under the name PPS, attract attention better than standard pictograms [45]. This is probably due in part to the color used in their pictograms.

We now return to the problematic phenomenon of information avoidance mentioned in the Introduction, whereby people avoid information, especially if the outcome might be undesirable. A consequence of this phenomenon is that PIs are no longer consulted. Pictograms might help to circumvent this problem as they attract attention and accordingly prompt the consumer to engage with them. This phenomenon has also been observed in the case of front-of-pack labels [46], which divert attention from nutrition facts panels. The consequence is that consumers rely on the information shown on front-of-pack labels. This is not detrimental if the labels are able to display the relevant information. Yet, one of the concerns about front-of-pack labels is that they could mislead consumers through overly simplified information content, for example, by using percentages [47]. However, in our pictogram study, we were able to show that the correct dosage could be selected when the appropriate pictograms were displayed, and they were recognized as relevant.

Why the position of the relevant warning pictogram had no effect is puzzling, especially since left-to-right directionality has also been observed in risk assessment [48], with the left being associated with “certain”, for instance. In the study by Kwak and Hüttel [48], participants were first presented—in the center of the screen—with the amount of money they would start the experiment with (e.g., USD 12). Depending on whether the game was framed as a “gain“ or a “loss”, they were then given a choice between a certain win or loss (e.g., USD 6) and a gamble with an uncertain outcome (50% chance of keeping the USD 12 and 50% chance of losing the USD 12). The authors found that the framing effect (i.e., the difference in the percentage of gamble choices between the gain and lose frame) was less pronounced when a certain option was visually placed on the left. Moreover, this effect was not exclusively due to the fact that the information presented on the left is processed with priority. Hence, they concluded that not only the position but also the *identity* of the information to be processed plays a role. For repetition, establishing the identity of an item or recognizing a difference in the identity of items requires mental processing (associated with effort; see also [49]). However, for these “left-to-right” position effects to occur, the items only need to be perceived as not identical; thus, they can be objectively the same (see Nisbett and Wilson’s study of stockings [50]). 

Yet, there are studies that do not show “left-to-right” position effects (or edge advantage effects) but the so-called edge aversion effects or centrality preferences (see Kim et al. [51]). Here, we exclude those studies that require a motor action, such as the study by Keller et al. [52], and/or those that do not require the processing of the items because they are identical or perceived to be identical; for example, see Christenfield’s study [53] of cereal boxes. An example of centrality preferences can be found in the work of Kim et al. [51], which showed that participants preferred wine bottles, tacos, etc., presented in the middle of a horizontal arrangement. Regardless of the observed effect (to date, it is still not clear why these different effects occur), the second position should be “special” (e.g., perceived as riskier) compared with the first. 

The only position effect (i.e., a difference between the first and second position) found in our study was related to the correct dosage choice when participants had consulted the PI. In this regard, fewer correct dosage decisions were made when the warning pictogram was presented in the first position (58%) than in the second position (77%; *z* = 2.51; *p* < 0.05; s. proposition test). That is, the values of participants who had consulted the PI versus those who had not were significantly lower in the first position (approximately—15%; *z* = 2.38; *p* < 0.05) and higher, but not significantly so, in the second position (approximately +7%; *z* = 1.21; *p* = 0.23). Since both groups had seen the same pictogram arrangement, reading the PI must have led to an adjustment of the previously formed mental value (i.e., an anchor adjustment process). 

For example, Marchiori et al. [54] showed that their participants estimated their potential food intake (e.g., grams of pasta that would be consumed in a restaurant) differently depending on the anchor previously presented (low: 75 g; high: 300 g). For example, participants presented with a low anchor estimated their potential food intake to be lower than a “no anchor” condition. Therefore, we can assume that our participants estimated the medication’s risk based on the pictogram arrangement (in which the position of the warning pictogram proved to be irrelevant), based on which the group that did *not* read the PI then made their dosage decision. The group that consulted the PI, on the other hand, made adjustments relative to a neutral reference point and did so differently depending on the position of the warning pictogram. For example, they tended to make a downward adjustment when the warning was in the first position. This led to fewer correct dosage decisions overall. However, these assumptions require further investigation, as it is unclear what may have caused this PI-induced adjustment.

Regarding possible study limitations, one such limitation could be that our study population consisted of students, which could potentially limit the generalizability of the study results (e.g., to non-academic, older individuals). Given that the majority of ZHAW students are second-degree students (i.e., most have already completed training in another profession) and accordingly have a greater diversity in terms of demographic data than students at other universities, we think this potential limitation is probably not applicable. Although there is a potential limitation regarding generalizability, it is worth testing students as a study population for a number of reasons. Students frequently consume OTCs (e.g., analgesics), for example, during exams (to improve concentration or to combat headaches). However, it is important to note that the number of students consuming OTCs varies widely from country to country (e.g., in Ethiopia, the rate is 43%, whereas in Saudi Arabia, which is known for its high OTC consumption, it is 85%; see also Sánchez-Sánchez et al. [55], according to which students are among the largest consumers of OTCs, especially analgesics, at 86%). It is further known from the study of Orayi et al. [32] that although students (85%) think that they know how to self-medicate, they are at the same time subject to the misconception (54%; mainly male students) that the overuse of OTC (such as Panadol Extra) is safe for a certain period of time. Therefore, in order to determine how to promote responsible usage of OTCs, it is worthwhile to examine students’ unhealthy patterns of medication intake (e.g., overdosing; see also Bekele et al. [56], indicating that about 24% of students double the dosage when it is not effective). Universities in particular should play a supporting role in highlighting the dangers of taking OTC medications (e.g., during study and exam periods). Another possible limitation of this study could be that the hypothetical scenarios used are too far removed from reality. However, we believe this should not be the case here since the last author has expertise in this area (first education: certified nurse, advanced diploma of higher education; with working experience in hospital and home-based care settings).

Despite the fact that this study has shown pictograms have positive effects on safe medication intake, further questions should be addressed before using pharmaceutical pictograms on medication packaging. For example, apart from testing them on other study populations, it should be investigated whether it makes a difference in which position the pictograms are displayed on the package (e.g., top left vs. top right or bottom left), or whether and how the number of pictograms influences risk perception and hence behavior. The last point is especially relevant since human memory capacity is limited. The more information people need to attend to the more difficult the task becomes; this has been shown in Reijnen et al. [45] regarding pictograms. The more that is known about the effect of pharmaceutical pictograms on medication packages, the more they can help to counteract non-adherence. However, the factors influencing medication adherence are many and varied. Multifaceted and tailored interventions are necessary to improve the self-administration of medication [57,58]. It might also be interesting to investigate whether a “read the package insert” pictogram could increase the frequency of PI consultations. In most cases, it is desirable that the individuals also inform themselves about the respective medication and its application by reading the PI [59,60].

## 5. Practice Implications

Placing the most important warnings or side effects on the package regarding safe medication use may be particularly useful in situations where additional professional help from a health worker (e.g., a pharmacist) is not available. This could be the case, for example, with pharmacy vending machines offering OTC medicines (see https://web.archive.org/web/20230817062229/, https://www.fm1today.ch/ostschweiz/appenzellerland/apotheken-automat-kommt-nach-herisau-135508151, accessed on 14 August 2023) where such a machine has been tested at a Swiss train station). However, pictograms could also be used in a more tailored way. For example, it could be interesting for health professionals to try applying pharmaceutical pictograms in the form of stickers to the package in an individual-specific way. Kelly and Sharot [61] have shown that there are individual differences in people’s information-seeking behavior. Depending on the type of person and their motives when seeking—or ignoring—information, the relevant facts are weighted differently. For example, an anxious person might use a different information-seeking strategy when pictograms on side effects are shown than an impulsive person. This is another aspect that should be considered regarding the personalization of pharmaceutical pictograms and adherence. 

## Figures and Tables

**Figure 1 behavsci-13-00696-f001:**
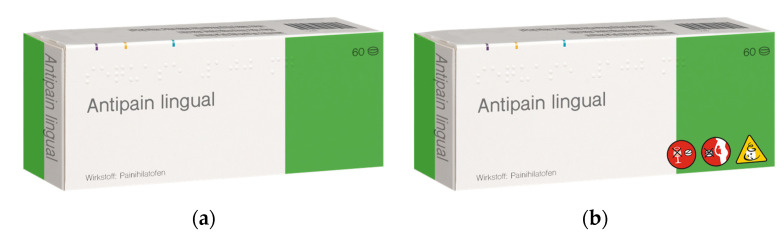
Example of a medication package used: (**a**) package without pictograms, respectively; (**b**) with pictograms. For every scenario, a different package design was used.

**Figure 2 behavsci-13-00696-f002:**
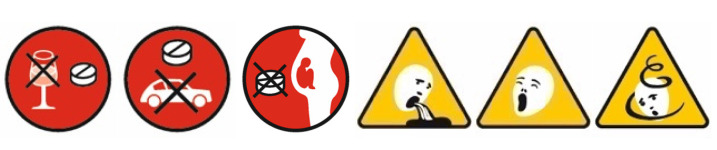
Pharmaceutical pictograms used: warning pictograms for “do not take in combination with alcohol”, “do not drive after taking”, and “do not take during pregnancy” and side effect pictograms for “can cause nausea”, “can cause tiredness”, and “can cause dizziness” (from left to right).

**Figure 3 behavsci-13-00696-f003:**
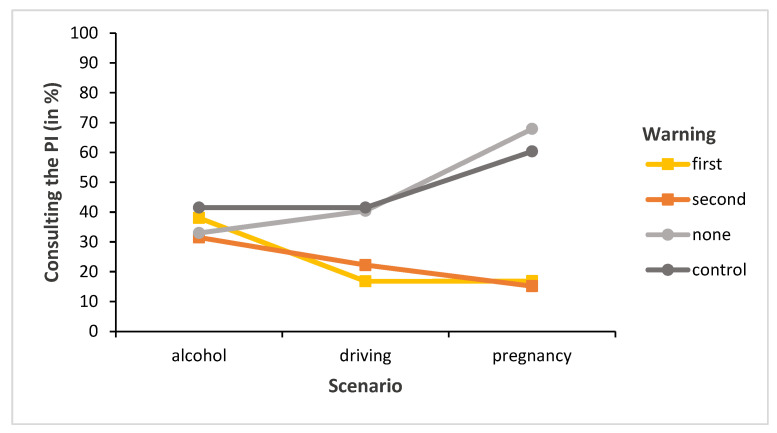
Frequency of PI consultation.

**Figure 4 behavsci-13-00696-f004:**
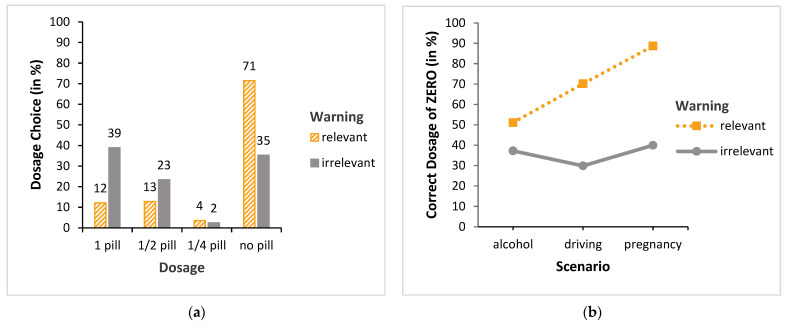
Dosage choices without PI consultation: (**a**) all dosage choices; (**b**) correct choice of taking “no pill”.

**Figure 5 behavsci-13-00696-f005:**
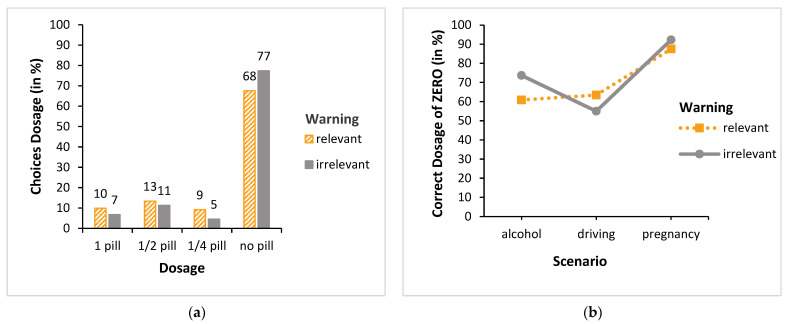
Dosage choices with PI consultation: (**a**) all dosage choices; (**b**) correct choice of taking “no pill”.

**Figure 6 behavsci-13-00696-f006:**
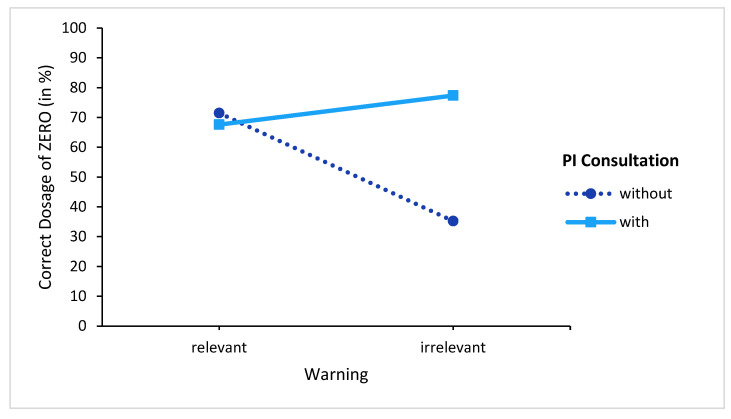
Dosage choices (summarized over the scenarios).

## Data Availability

Aggregated data are available upon request to the corresponding author.

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
