# Peer review of "Do Pictograms on Medication Packages Cause People to Consult Package Inserts Less Often? If so, With What Consequences?"

_behavsci, 2023, doi:10.3390/bs13080696_

Round 1

Reviewer 1 Report

[Behavioral Sciences] Manuscript ID: behavsci-2515736 - Review Report

Article

Do Pictograms on Medication Packages Cause People to Consult Package Inserts Less Often? If so, With What Consequences?

By: Ester Reijnen, Lea Laasner Vogt, Swen J. Kühne and Jan P. Fiechter,

Manuscript Summary:

The study is about pharmaceutical pictograms warnings shown on a medication package, and how they are an important aid to improve adherence. The study has involved participants (358) were presented with 3 fictitious scenarios. The study has concluded with the following:

Pictograms generally played no role either in review of the PI consultation or dosage decisions in the alcohol scenario. Providing warning-relevant pictograms on medication packages can help people know when they should not take a medication even without reading the PI.

General comments:

-          Very important study. I do encourage to publish its results. But still, it needs some improvements in terms of presenting the deep analysis of the study.  This was not done.

-          The study has been based on statistical analysis for data collected from students (this is an issue).

-          I was worried about the length of the article, … its is a bit short to be a full journal article.

more comments:

-          The abstract is not well written.  It needs to be made much focused towards the works done.

-          The previous and literature work is not well expressed and presented.  This improvement to show and indicate to the most important previous work and how this article results are different. 

-          The study has indicated to (decision making).  I did not found material related to (decision making).

-          There is a major issue with the involved participants (the 358).  All are students, … which means that all are well and healthy, …  whereas such (Pictograms on Medication Packages) should be more directed towards the public (oldy, teens, and (including students)).

-          Data information on the figure (Figure 6. Choices dosage overall. 274) needs an improvement.

-          The conclusion is not well written also.  It needs to be made much focused towards the works done.

-          For the (References), all the used references (47) are fine.

-          I found few English writing stated that they some improvements,

-          English writing of the article is needed to be checked .. examples are ..

Pictograms might help to circumvent this problem as they attract 325 attention.

Why might the use of pictograms on medication packages be beneficial? In this re-28

Reviewer 2 Report

This well-written study is focusing on the impact of pictograms on consulting package inserts. However, some improvements need to be made before it is ready for publication.

Please add your hypotheses to the introduction section.

-          In the method section:

How did you recruit the participants? How the study was presented to them? Please add this info to the method section.

Please add info on the participants’ education level to the method or result section.

Do you have info on medical history of the participants? Is it possible that people with a more extensive medical history may yield different results compared to those with no past medical history in this study? Please discuss about this in the limitation section.

“Participants were able to enter an iPad raffle or, students from the school of Applied Psychology, could receive course credit for their participation instead (which 10% did).” Please provide info on the percentage of individuals who were from applied psychology department in the result section of the manuscript.

-          In the discussion section:

The authors say, “First, we tested our predictions on a sample of students. However, since the majority of ZHAW students are pursuing a second degree (i.e., most of them have already completed training in another profession), they exhibit a great deal of diversity in demographic data such as age (see above).” This is an important limitation of this study. The results may not be generalizable to the broader population, particularly to individuals with different age, education level, and other demographic characteristics. The authors should further elaborate on this limitation in the manuscript's discussion section.

Reviewer 3 Report

I admire your work in the current study but have several concerns. Would you please check the comments?

A brief summary

This research was conducted to investigate whether the fictitious scenarios and the arrangement of pharmaceutical pictograms influenced participants to consult the package insert and correctly determine the appropriate dosage. Participants were shown three scenarios in which they needed to take a painkiller. In each case, they were presented with a medication package that could contain three possible pictogram selections or arrangements. As a result, participants consulted the package insert less in the driving and pregnancy scenarios regardless of the order of relevant pictograms. In addition, participants could determine the appropriate dosage when the relevant pictogram was on the package, revealing that even when participants did not consult the package insert, a relevant pictogram can help inform the appropriate use of the medication.

General concept comments

It is crucial to provide precise and easily understandable dosage information because misunderstandings about medication can be fatal. Hence, I think the objectives of this study, which aim to investigate the influence of the scenarios and the arrangements of pictogram on PI consultations, are convincing. In addition, the results were clear.

However, while the interpretation presented in the discussion section is convincing, additional explanation is needed to clarify why the hypothesis that the arrangement of the pictograms would influence PI consultation was not supported. Hence, I think that the variables in the warning factor should not be collapsed into relevant/irrelevant pictogram conditions, as explained in specific comments section.

    In addition, the Materials and Methods section contains some ambiguous points that require further explanation to improve the understanding of this research.

Specific comments

Introduction

p. 2, l. 84 – 88.

1)    In the manuscript, it is said that “The prerequisite for this is that people recognize whether the pictograms displayed are relevant for them in the respective situation or not.”, however, I am having difficulty understanding why recognizing the relevance of the pictograms is a prerequisite for the process described in the manuscript. Could you please provide more context or explanation? After seeing the pictogram or consulting the PI, participants are able to recognize whether the pictogram is relevant to themselves and their situations. Hence, why is it necessary to recognize the relevance of the pictogram beforehand?

2)    In addition, considering the example provided in the manuscript, it may be worth exploring whether the gender of the participants had any influence on the results, particularly in the pregnancy scenario. While both male and female participants can imagine the pregnancy situation, it is possible that the level of reality in their imagination may differ. Could you please provide more information on whether gender was considered as a potential factor in the analysis?

p. 2, l. 89 – 100.

 3)    In the manuscript, it is stated that “A secondary question is whether the position or the sequence of pictograms displayed matters”. Based on this, I assumed that the position of the pictogram was considered a crucial factor. However, this factor is not mentioned much in the discussion section, so it is unclear why the sequence of pictograms did not play a role in the dosage decision and the PI consultation. Therefore, it would be helpful to provide an interpretation for this.

4)    To provide this interpretation, it is supposed that analyzing the warning conditions using separated data, as shown in Figure 3, is more appropriate than using the collapsed data. I understand that it is important to emphasize the difference between the relevant pictograms and irrelevant one. However, even if separated data were used, it is still possible to emphasize the effect of the relevance. At the same time, it is also possible to show that there is no significant effect of the arrangement of the pictograms by using separated data. Based on Figure 3 and its analysis, there was no significant difference between the first and second conditions, so it is possible to interpret that the effect of the arrangement was not significant. On the other hand, at the same time, it appears that there is a significant difference between the relevant conditions and irrelevant conditions in both driving and pregnancy scenarios, so it is possible to interpret that the relevancy of the pictogram is the important factor in the specific situation. Like this, analysis using separated data can provide a richer interpretation compared to using collapsed data, without compromising the interpretation for the importance of relevance. Therefore, I believe that separated data should be utilized in other analyses as well.

5)    By using this method, it is possible to claim that the contents of the pictogram are more important than its location, so I think that this is an important discovery of this study.

p. 3, l. 101 – 105.

6)    As mentioned earlier, it was somewhat difficult to understand why relevancy of the pictogram is an important factor for this study. Could you provide more information?

Materials and Methods

7)    Participants, Stimulus material and Procedure and Design sections are now indented. However, I believe that they should be treated as subsections, such as 2.1, 2.2, and 2.3, for better organization.

Participants

8)    According to the Introduction section, the pictograms are sensitive to culture. In the present study, were the participants asked about their cultural backgrounds? If so, how diverse were the cultural backgrounds? Additionally, did you check whether all participants could understand the intended meaning of each pictogram?

Procedure and design

9)    In this online study, I imagine that participants were sequentially presented with the scenario sentence, an image of the medication package, the PI (if requested), and the answer sheet for dosage. However, it is somewhat unclear how the stimuli were displayed and the sequence in which they were presented. To better understand the experimental flow, would it be possible to add a figure to the Procedure section that illustrates the order in which participants were presented with the scenario sentence, image of the medication package, PI, and answer sheet for dosage?

p. 4, l. 142– 148

10)    As mentioned earlier in this comment, I assumed that the position of the pictogram was considered a crucial factor. However, I noticed that the 'third position' condition was not included. Can you explain why this was the case?

11)    According to the manuscript, the first position condition had two side effect pictograms and one warning pictogram. On the other hand, the second position condition had one side effect pictogram and two warning pictograms. It is supposed that the same pictograms could be used by simply changing the order of the warning pictogram (e.g., first position condition, Warning A, Warning B, Side effect C; second position condition, Warning B, Warning A, Side effect C). Could you please explain why the number of warning and side effect pictograms differed between the conditions?

p. 4, l. 149– 156

12)    I find it somewhat difficult to understand the control (4th) condition. My interpretation is that in the control group, participants were presented with a package shown in Figure 1 (a) in all three scenarios. Additionally, participants who were presented with packages with pictograms joined the three warning conditions (first, second, and none), and participants assigned to the control group were different individuals. Is my interpretation correct? If this is the case, would you clarify how many participants were assigned to each group?

13)    According to the manuscript, each participant was presented with all three scenarios. Therefore, it is possible that participants might have consulted the package insert in the first trial and checked it again in the next or last trial. In this case, participants might have noticed that the PI was identical across all conditions. This fact could have influenced their dosage decision. Were there participants who checked the package insert in this manner? If so, could you explain whether this case influenced the results?

p. 4, l. 157– 164

14)    The manuscript states that the 'second dependent variable' is 'dosage', but the first dependent variable is ambiguous. Could you please clarify what the first dependent variable is?

15)    Based on the information provided in this paragraph, the dosage options for the medication were 1 pill, ½ pill, ¼ pill, and no pill. I am curious to know if these dosage options are common in the participants' culture. In my culture, the recommended dosage of medication is often an integer number, such as one or two pills. If this experiment were conducted in my country, it is possible that participants would hardly choose ½ pill or ¼ pill due to cultural differences in medication practices. I would appreciate if the authors could provide further explanation on the rationale behind these dosage options.

16)    Could participants notice the recommended dosage (i.e., 1 pill) solely based on the information provided on the package? In other words, was the dosage information included on the package? If there was no dosage information on the package, how did participants who did not consult the PI decide on their dosage?

17)    When and how was each participant offered the option of consulting the PI? Could you provide more information about the timing of the offer and how the PI was displayed?

Results

18)    In my previous comment, I expressed concern about collapsing the variables in the warning factor into relevant/irrelevant pictogram conditions. I would appreciate it if you could take another look at this.

p. 4, l. 171– 180

19)    Why was the number of participants lower in the control group? Additionally, could you please explain why a probit regression model was not possible for the control group due to the lower number of participants? Given that the dependent variable was binary, I believe it may be possible to use a probit regression model for the control group.

20)    In the Participants section, it is stated that there were 358 participants. Additionally, in the Results section, it is mentioned that 3% of the participants were excluded from the analysis. On the other hand, in the Note section, it is stated that 358 participants who made 3 decisions were included in the analysis. Since these numbers seem to conflict with each other, could you please clarify which one is correct?

21)    According to the analysis in this paragraph, the interaction between the scenario and warning factors was significant. Therefore, please provide the information of the simple main effects and multiple comparisons.

Discussion

22)    In my previous comment, I expressed that the reason why the arrangement of the pictograms did not play a role should be mentioned. Please add the paragraph for this to Discussion section.

p. 9, l. 311– 317

23)    It might be somewhat difficult for readers to interpret meanings of “the earlier process” and “the later stage”. In my interpretation, the earlier process may mean the perception of the pictogram, and the later stages may mean cognition or decision-making stage, however, I am not confident that this interpretation is correct. Could you explain more in detail?

Conclusion

24)    The contents described in this section could be considered as 'Future prospects' rather than 'Conclusion.' Therefore, I suggest moving these sentences to the Discussion or Practice Implications sections and including a summary of the present study's findings here.

Practice Implication

p. 10, l. 370– 373

25)    In this sentence, an anxious person is compared to an impulsive person. However, anxiety and impulsiveness are different measures, so it is more appropriate to compare highly anxious (impulsive) individuals with low anxious (impulsive) ones.

Reference

26)    Reference No. 15 shows that 58.8% of studies reported a significant effect of the pictogram, but this manuscript uses 58% instead. Rounding the figure to 59% would be appropriate.

27)    Reference No. 29 reported that more than half (51.5 %) participants read the information leaflet in the drug package. This means that persons who read the leaflet was a majority, at least, they were not “only a few people” as mentioned in l. 281 - 284. Additionally, I believe that this percentage would not be “similar” to the number described in this study (34%). While I understand that the fact that only approximately half of the individuals read the leaflet is still concerning, could you please consider using a softer expression in the manuscript?

Round 2

Reviewer 2 Report

The authors have addressed my comments/suggestions.

Reviewer 3 Report

I appreciate your sincere efforts in addressing my review and found that the manuscript was improved significantly. Would you please check the comments below?

1)  Would you add “Example of an PI (Figure 1)”, “Pictogram Combination Used per Scenario ~ (Table 4)”, and “The numbers of participants (shown in the answer for the comment 12)” to the supplementary material? It would help readers understand the method of the present study.

2)  I apologize if comment 20 was confusing. In my interpretation, the number of participants included in the analysis can be calculated as follows: 358 (the total number of participants) minus 11 (3% of 358, the number of excluded participants) equals 347 (the actual number of participants included in the analysis). Therefore, the total number of decisions would be 347 multiplied by 3, which equals 1041. Alternatively, if 358 was the number of participants included in the analysis, then the total number of participants should have been 358 divided by (1 - 0.03), resulting in 369 (the total number of participants). This is why I found the numbers to be conflicting. Could you please clarify the total number of participants and the number of excluded ones?

3) Please correct the section number of Practice Implications from 6 to 5.
